# Landau theory for non-equilibrium steady states

## Camille Aron[1,2] and Claudio Chamon[3]

**1** Laboratoire de Physique, École Normale Supérieure, CNRS, Université PSL,
Sorbonne Université, Université de Paris, 75005 Paris, France
**2** Instituut voor Theoretische Fysica, KU Leuven, Belgium
**3** Physics Department, Boston University, Boston, Massachusetts 02215, USA

## Abstract

We examine how non-equilibrium steady states close to a continuous phase transition can still be described by a Landau potential if one forgoes the assumption of analyticity. In a system simultaneously coupled to several baths at different temperatures, the non-analytic potential arises from the different density of states of the baths. In periodically driven-dissipative systems, the role of multiple baths is played by a single bath transferring energy at different harmonics of the driving frequency. The mean-field critical exponents become dependent on the low-energy features of the two most singular baths. We propose an extension beyond mean field.



# 1 Introduction

The Landau-Ginzburg theory of equilibrium phase transitions builds on simple principles, namely symmetry, locality, and analyticity of the free energy potential. These simple assumptions, when fed into the Renormalization Group framework, lead to universality of the critical exponents at second-order phase transitions, which depend only on the specific symmetry and the dimensionality of space.

Systems that are not at equilibrium, on the other hand, are often thought to behave each in its own different, non-universal way, and are thus studied under a hodgepodge of theoretical techniques. Our goal in this paper is to salvage whichever piece of universality is possible in those non-equilibrium systems that reach a steady state. In those cases, one can extract from the probability distribution of the system's state a potential that parallels the Landau-Ginzburg free energy at equilibrium. We look into Landau theory, and examine the assumptions that one must forgo when the steady state is not an equilibrium one.

Non-equilibrium phase transitions have been intensely investigated in the context of the so-called driven-diffusive systems [1, 2], in which the dynamics conserves a global quantity such as the particle number. A notable instance is the driven lattice gas [3, 4], where classical non-overlapping particles hop to unoccupied neighboring sites with rates that depend on an external uniform electric field. There, the field-theoretic approaches mostly concentrated on the mesoscopic dynamics by proposing equations of motion of the Langevin type, or their associated Martin-Siggia-Rose-Janssen-deDominicis action, to generalize the Model B [5] to non-equilibrium situations.

Another class of non-equilibrium systems are the so-called driven-dissipative systems, with no conserved quantity. There is continued interest in the study of growth processes, such as the directed percolation [2, 6, 7] or the Kardar-Parisi-Zhang problems [8, 9]. Similarly to the driven-diffusive systems, space plays a fundamental role in their non-equilibrium nature in the sense that these models need a formulation in at least one spatial dimension in order to display non-Gibbsian stationary states.

In this manuscript, we study perhaps an even simpler class of driven-dissipative systems: those for which the stationary states are expected to be homogeneous and isotropic. Their appeal is the relative simplicity in which to examine basic yet fundamental questions (here the prospect of a Landau theory for the non-equilibrium steady states via single-site mean-field methods), in contrast to the driven-diffusive systems which typically exhibit directional currents and possibly phase separation and thus require more sophisticated approaches.

We focus on the $\mathbb{Z}_2$-symmetric magnet, *i.e.* the Ising model, driven to a uniform non-equilibrium steady state (NESS) by either multiple baths at different temperatures, or by a fast periodic longitudinal magnetic field. It was previously analytically argued [10], numerically confirmed in many instances, and generally believed, that the related continuous ferromagnetic transitions still belong to the *equilibrium* Ising universality class. The argument was based on both mean-field and finite-dimensional computations. The latter consisted in showing that even when the microscopic dynamics do not derive from a potential, an RG procedure washes away non-potential forces and Model A dynamics with the ordinary $\varphi^4$ potential are recovered at large scales. Noteworthy, these computations relied on the assumption that those non-potential forces are analytic in $\varphi$. Our main result consists in showing via a mean-field approach that the Landau potential of non-equilibrium steady states can in fact feature non-analytic terms, reading

$$\mathcal{V}_{\text{NESS}}(\varphi) = \underbrace{a_2\varphi^2 + a_4\varphi^4}_{\text{analytic}} + \underbrace{c_\alpha|\varphi|^{2+\alpha}}_{\text{non-analytic}} \,, \tag{1}$$

where the additional term to the ordinary $\varphi^4$ potential is a signature of the non-equilibrium

nature of the steady state. The exponent $\alpha > 0$ has its origin in the low-energy spectrum of the environment and can be non-integer valued. The coefficients $a_2$, $a_4$, and $c_\alpha$, are smooth functions of the external parameters and $c_\alpha$ vanishes at equilibrium.

The additional non-analytic term in Eq. (1) alters the phase transition and the static critical exponents of the ordinary $\varphi^4$ theory. This departs from the equilibrium classes of universality in that the critical exponents now also depend on the low-energy behavior of the environment's density of states.

The paper is organized as follows. In Sec. 2 we briefly review the assumptions of Landau-Ginzburg theory at equilibrium, and how to extract the ordinary $\varphi^4$ potential from the spin dynamics when detailed balance holds. In Sec. 3 we identify the building principles of a Landau theory for non-equilibrium steady states, while in Sec. 4 we exemplify our non-equilibrium theory on two different types of driven-dissipative Ising models: one coupled to baths at different temperatures, and one under time-periodic driving. We close in Sec. 5 by proposing a Landau-Ginzburg free energy for the non-equilibrium steady states in finite dimensions, along with a discussion of the underlying assumptions and the possible difficulties in carrying out an RG calculation with it as starting point.

## 2 Brief review of Landau-Ginzburg theory at equilibrium

### 2.1 Landau-Ginzburg free energy: building principles

When seeking an effective-field-theory description of a many-body system, attempting a derivation starting from the microscopics is typically an unsurmountable task. More often than not, the underlying microscopic degrees of freedom are plentiful, and their quantitative modeling is unknown. Moreover, tracing over those degrees of freedom may be unfeasible, especially if they are interacting. Therefore, one must rely on general arguments to come up with a free-energy functional $\mathcal{F}[\varphi]$ which describes the probability distribution

$$P[\varphi] \sim \mathrm{e}^{-\mathcal{F}[\varphi]} \tag{2}$$

of configurations $\varphi(x)$ of the order parameter. For simplicity, we assume here and throughout this manuscript that the order parameter of interest, $\varphi(x)$, is a scalar. At thermal equilibrium and close to a second-order phase transition, the Landau-Ginzburg's approach consists in considering the most generic expression of $\mathcal{F}[\varphi]$ that satisfies the following principles (see, *e.g.*, Ref. [11]):

- Locality: $\mathcal{F}[\varphi] = \int \mathrm{d}x \; \mathcal{L}(\varphi, \nabla\varphi, \dots; x)$. $\mathcal{F}$ can be expressed in terms of a local free-energy density $\mathcal{L}$.

- Symmetries: $\mathcal{F}[\varphi] = \mathcal{F}[S\varphi]$, up to boundary terms. The Landau-Ginzburg free-energy must comply with all the symmetries, global and local, of the order parameter. For example, the global $\mathbb{Z}_2$ symmetry of the Ising model imposes $\mathcal{L}$ to be invariant under $S : \varphi(x) \mapsto -\varphi(x)$. As another example, if the system is statistically invariant under translations, $\mathcal{L}$ does not depend explicitly on $x$.

- Analyticity: $\mathcal{L}(\varphi, \nabla\varphi, \dots) = a_1\varphi + a_2\varphi^2 + b_2(\nabla\varphi)^2 + \dots$. $\mathcal{L}$ is assumed to be analytic in the field $\varphi$ and its derivatives. This assumption is usually justified in the literature by arguing that any non-analyticity present at a microscopic level is expected to be washed out at a more mesoscopic level, after the corresponding degrees of freedom have been traced out.

- Smoothness of parameters: the coefficients $a_1$, $a_2$, $b_2$ ... are assumed to be smooth and continuous non-universal functions of the external parameters (temperature, pressure, etc.)

- Stability: $\int \mathcal{D}[\varphi] e^{-\mathcal{F}[\varphi]} < \infty$. For the probability distribution to be well defined, the largest power of $\varphi$ must be even and its coefficient positive.

- RG relevance: $\mathcal{L}(\varphi, \nabla\varphi, \ldots)$ is defined up to terms which are irrelevant in an RG sense. For example, the terms of order $\varphi^6$ and higher are irrelevant to a $\varphi^4$ theory in $4 - \epsilon$ dimensions and above.

These principles were given solid foundations by the Renormalization Group (RG) theory. In particular, the RG theory taught us that the parameters $a_1$, $a_2$, $b_2$ ..., depend and flow with the scale at which the system is probed. Low energy physics and critical physics are controlled by fixed points of the RG flow and their stability.

## 2.2 $\varphi^4$ theory from the dissipative Ising model

In the pursuit of identifying the effective field theory that correctly describes an extended many-body system, it has often proven useful to first address the problem within a mean-field picture. The mean-field approximation consists in neglecting possible spatial fluctuations of the order parameter, *i.e.* working with uniform configurations $\varphi(x) = \varphi$. There, the identification of the Landau-Ginzburg free energy boils down to the identification of an effective potential: $\mathcal{L}(\varphi, \nabla\varphi = 0) = \mathcal{V}(\varphi)$. Later, once the mean-field description is well under control, spatial fluctuations can be re-incorporated in the theory and their effect methodically studied.

This is precisely the approach we shall follow in this manuscript, working in the context of the notorious Ising model whose *equilibrium* effective field theory is the well-known $O(n = 1)$-symmetric $\varphi^4$ theory. To better prepare the ensuing non-equilibrium discussions, we briefly review the mean-field derivation of the later from the perspective of its equilibrium dynamics.

**Dissipative Ising model**   Let us consider the equilibrium dynamics of the dissipative Ising model, *i.e.* the Ising model coupled to a simple thermal environment. The Ising Hamiltonian reads

$$H = -\frac{J}{z} \sum_{\langle ij \rangle} S_i^z S_j^z \,, \tag{3}$$

where each spin $S_i^z = \pm 1$ is ferromagnetically coupled to its $z$ nearest neighbors. Below, we take the ferromagnetic coupling $J > 0$ as the unit of energy by setting $J := 1$. The environment is assumed to be a collection of identical thermal reservoirs at temperature $T \equiv \beta^{-1}$ that are locally and weakly coupled to the spins. This model is often referred as the kinetic Ising model [12]. The non-conserved order parameter of interest is naturally the average magnetization $\varphi \equiv \langle S_i^z \rangle$. In two dimensions and above, this model is well known to exhibit a finite-temperature second-order phase transition between a $\mathbb{Z}_2$-symmetric paramagnetic phase ($\varphi = 0$) and a $\mathbb{Z}_2$-broken ferromagnetic phase ($\varphi \neq 0$).

**Single-spin mean-field description**   At the level of the Ising spins, we implement the mean-field approximation by considering an auxiliary single-spin impurity problem. It consists of a single spin subject to a coherent Weiss field $h_W$ created by the neighboring spins, and to incoherent thermal spin flips –the rates of which obey detailed balance– created by the local environment at equilibrium (EQ). The self consistency (SC) between the original dissipative

Ising model and the impurity problem is achieved by imposing the same average magnetization $\varphi$ in both models and the Weiss field $h_{\mathrm{W}}(\varphi) \overset{\mathrm{SC}}{=} \varphi$.

The dynamics of the mean-field order parameter $\varphi$ may be simply written as rate equations on the probabilities $P_{\downarrow} = \frac{1-\varphi}{2}$ and $P_{\uparrow} = \frac{1+\varphi}{2}$ for the impurity spin to be down or up, respectively:

$$\partial_t P_{\uparrow} = P_{\downarrow} R_{\downarrow\uparrow} - P_{\uparrow} R_{\uparrow\downarrow}, \tag{4}$$

with the constraint $P_{\uparrow} + P_{\downarrow} = 1$. $R_{\downarrow\uparrow}$ and $R_{\uparrow\downarrow}$ are the rates of flipping the impurity spin up or down, respectively. They depend of the local Weiss field $h_{\mathrm{W}}$. Once a steady state is reached, *i.e.* $\partial_t P_{\uparrow} = 0$, the self-consistency equation on the mean-field order parameter reads

$$\varphi \overset{\mathrm{SC}}{=} \hat{R}/R(\varphi) \overset{\mathrm{EQ}}{=} \tanh(\beta\varphi), \tag{5}$$

where we introduced $\hat{R} \equiv R_{\downarrow\uparrow} - R_{\uparrow\downarrow}$ and $R \equiv R_{\downarrow\uparrow} + R_{\uparrow\downarrow}$. This ratio of rates, $\hat{R}/R$, is a central object to this manuscript: it dictates the single-spin dynamics. In the last step, we made use of the detailed balance condition, $\hat{R}/R \overset{\mathrm{EQ}}{=} \tanh(\beta h_{\mathrm{W}})$, which is a signature of the equilibrium nature of the environment. Below, when dealing with non-equilibrium steady states, we shall relax this condition.

**Landau potential**  The solutions of the self-consistency equation (5) can be recast as the extrema of the effective potential $\mathcal{V}_{\mathrm{EQ}}(\varphi)$ defined as

$$\mathcal{V}_{\mathrm{EQ}}(\varphi) \equiv \int^{\varphi} \mathrm{d}\varphi \, \frac{\varphi - \tanh(\beta\varphi)}{D(\varphi)}. \tag{6}$$

The denominator $D(\varphi)$ is present to accommodate equivalent re-writings of Eq. (5). In App. A, we show that $D(\varphi)$ is a well-behaved positive and even function, the precise choice of which is inconsequential to the resulting theory. To simply give the reader a flavor of this statement, we compare the effective potentials that result from two different choices for $D(\varphi)$. If one chooses $D(\varphi) := 1$, one obtains the effective potential

$$\mathcal{V}_{\mathrm{EQ}}(\varphi) = \frac{1}{2}(1-\beta)\varphi^2 + \frac{1}{12}\beta^3\varphi^4 + \mathcal{O}(\varphi^6), \tag{7}$$

whereas another choice of interest for the next Section, namely $D(\varphi) := 1 - \varphi \tanh(\beta\varphi)$, yields

$$\mathcal{V}_{\mathrm{EQ}}(\varphi) = \frac{1}{2}(1-\beta)\varphi^2 + \frac{1}{12}[\beta^3 + 3\beta(1-\beta)]\varphi^4 + \mathcal{O}(\varphi^6). \tag{8}$$

It is clear that these two choices of $D(\varphi)$ predict the same physics: a second-order phase transition at the critical temperature $T_c = 1$. The difference in the coefficients of the $\varphi^4$ terms does not affect the nature of the symmetry-breaking mechanism, vanishes at criticality, and will be washed out after a few RG steps away from criticality. Note that these two potentials are related by a smooth change of variable: $\varphi \mapsto \varphi + \frac{1}{4}\beta\varphi^3$.

**Landau-Ginzburg free-energy**  To depart from the mean-field picture, one upgrades $\varphi$ to a fluctuating quantity $\varphi(x)$ and proposes the following Landau-Ginzburg free-energy, sometimes referred as the Landau-Ginzburg-Wilson Hamiltonian,

$$\mathcal{F}_{\mathrm{EQ}}[\varphi] = \int \mathrm{d}x \, \frac{1}{2}(\nabla\varphi)^2 + \mathcal{V}_{\mathrm{EQ}}(\varphi), \tag{9}$$

where the dispersive term is the only gradient term allowed by the principles listed above. One obtains the expected $\varphi^4$ field theory which naturally boils down to the mean-field theory for uniform configurations $\varphi(x) = \varphi \; \forall x$.

# 3 Landau potential for non-equilibrium steady states

We now move away from thermal equilibrium, and aim at identifying the building principles of a Landau-Ginzburg theory for the non-equilibrium steady states (NESS). By non-equilibrium steady states, we have in mind states that are non-thermal but that are invariant under *infinitesimal* time translations. We shall see in Sec. 4.2 that under certain conditions, the case of time-periodic states can also be described by a static Landau theory.

Once a system with a fluctuating local order parameter $\varphi(x, t)$ has reached a stable non-equilibrium steady state, there exists a stationary probability distribution

$$P_{\text{NESS}}[\varphi] \sim e^{-\mathcal{F}_{\text{NESS}}[\varphi]}, \tag{10}$$

which quantifies the statistical occurrence of configurations of the field $\varphi(x)$. Our objective is to lay out the principles that govern the expressions of the corresponding Landau-Ginzburg effective free energies, $\mathcal{F}_{\text{NESS}}[\varphi]$, close to a continuous phase transition between a disordered ($\varphi = 0$) and an ordered ($\varphi \neq 0$) phase.

Similarly to the equilibrium case reviewed in Sec. 2, the non-equilibrium steady-state construction will be based on the principles of locality, symmetry, stability, and smoothness of the parameters. However, the assumption of analyticity of the free-energy density will need to be abandoned.

We first focus on the potential part of the free-energy, $\mathcal{V}_{\text{NESS}}(\varphi)$, by working at the mean-field level. The addition of fluctuations on top of the mean-field picture will be discussed subsequently in Sec. 5. Using concrete examples, we shall show that $\mathcal{V}_{\text{NESS}}(\varphi)$ can feature non-analytic terms consistent with the overall $\mathbb{Z}_2$ symmetry, of the type

$$\mathcal{V}_{\text{NESS}}(\varphi) = \underbrace{a_2 \varphi^2 + a_4 \varphi^4}_{\text{analytic}} + \underbrace{c_\alpha |\varphi|^{2+\alpha}}_{\text{non-analytic}}, \tag{11}$$

where $\alpha > 0$ and the coefficient $c_\alpha$ is a smooth function of the external parameters that vanishes at equilibrium. Several of these non-analytic terms can be simultaneously present (see, *e.g.*, Sec. 4.2). This generic structure of the effective potential in non-equilibrium steady states is one the main results of this manuscript.

## 3.1 Single-spin mean-field description

Let us consider the Ising model in Eq. (3), but now subject to a non-equilibrium drive and to dissipation. The precise details do not matter as long as the non-equilibrium drive and the dissipation occur uniformly and *locally* on the spins. Let us furthermore assume that, after a transient, the system has reached a homogeneous isotropic non-equilibrium steady state. This will guarantee the validity of a single-site mean-field approach. Obviously, not all driven-dissipative conditions are compatible with the system reaching a non-equilibrium steady state. However, it is a reasonable assumption in the presence of DC drives, such as a constant temperature bias in the environment, and at a safe distance from any dynamical instability. Furthermore, even with AC drives, constant non-equilibrium steady states may still be recovered in a stroboscopic sense through a Floquet description of the periodic dynamics, as we shall exemplify in Sec. 4.2.

Similarly to what was done in equilibrium in Sec. 2.2, the dynamics may be treated within a single-spin self-consistent mean-field approximation. The equation (4) and the first equality in Eq. (5) still apply to a non-equilibrium scenario, and we obtain the self-consistency (SC) equation

$$\varphi \overset{\text{SC}}{=} \hat{R}/R(\varphi), \tag{12}$$

where the dynamical ratio $\hat{R}/R(\varphi)$ was defined below Eq. (5) in terms of the spin-flip rates. Here, given the non-equilibrium nature of the steady state, the ratio $\hat{R}/R(\varphi)$ does not obey the detailed balance condition and must therefore be computed explicitly from the system-bath dynamics. We can now readily generalize the definition of the effective potential made in Eq. (6) to non-equilibrium steady-state situations via

$$\mathcal{V}_{\text{NESS}}(\varphi) \equiv \int^{\varphi} \mathrm{d}\varphi \, \frac{\varphi - \hat{R}/R(\varphi)}{D(\varphi)} \, , \tag{13}$$

such that the extrema of $\mathcal{V}_{\text{NESS}}(\varphi)$ correspond to the solutions of Eq. (12). $D(\varphi)$ is a well-behaved positive and even function. We show in App. A that the precise choice of $D(\varphi)$ is inconsequential. It is noteworthy to remark that the above definition of $\mathcal{V}_{\text{NESS}}(\varphi)$ is "universal" in the sense that it only involves the dynamical quantity $\hat{R}/R$ and does not explicitly depend on the details of the model.

## 3.2 Finite-size fully-connected model

Here, we propose an alternative route towards a consistent definition of the mean-field effective potential $\mathcal{V}_{\text{NESS}}(\varphi)$ without any guess work, corroborating the definition proposed in Eq. (13).

Let us consider a fully-connected version of the driven-dissipative Ising model that we considered in Sec. 3.1. The system Hamiltonian reads

$$H = -\frac{1}{N} \sum_{ij} S_i^z S_j^z \, , \tag{14}$$

where the sum now runs over all pairs of spin, irrespective of their relative distance, and we assume here again that the non-equilibrium environment is uniform and acts locally on the spins. We follow the dynamics of the mean magnetization, $\varphi \equiv \frac{1}{N} \sum_{i=1}^{N} S_i^z$, when the total number of spins $N$ is large but finite. It is a stochastic process in which the random jumps are due to individual spin flips driven by the system-bath interaction. In App. C, we show that the dynamics of the probability distribution, $P(\varphi, t)$, obey the following Fokker-Planck equation

$$\partial_t P + \partial_\varphi J = 0 \, , \tag{15}$$

with the current density

$$J \equiv PR(\hat{R}/R - \varphi) - \frac{1}{N} \partial_\varphi \left[ PR(1 - \varphi\hat{R}/R) \right] + \mathcal{O}(1/N^2) \, . \tag{16}$$

The steady-state distribution $P_{\text{NESS}}(\varphi)$ is solution of $\partial_t P(\varphi, t) = 0$ and can be solved by finding the distribution with a null current $J = 0$ [1]. We obtain the stationary measure

$$P_{\text{NESS}}(\varphi) \sim \frac{1}{R(1 - \varphi\hat{R}/R)} e^{-N \int^{\varphi} \mathrm{d}\varphi \, \frac{\varphi - \hat{R}/R}{1 - \varphi\hat{R}/R}} \, . \tag{17}$$

Discarding the factors which are sub-leading in $N$, we obtain the following definition of the effective potential

$$\mathcal{V}_{\text{NESS}}(\varphi) \equiv \int^{\varphi} \mathrm{d}\varphi \, \frac{\varphi - \hat{R}/R(\varphi)}{D(\varphi)} \, , \tag{18}$$

---

[1] It is easy to show that in the limit $N \to \infty$ where the distribution is peaked on the solutions of $\varphi = \hat{R}/R$, i.e. $P(\varphi) \sim \delta(\varphi - \hat{R}/R)$, there are no solutions with a non vanishing current $J \neq 0$.

where

$$D(\varphi) = 1 - \varphi \hat{R}/R(\varphi) \,. \tag{19}$$

Naturally, this is consistent with the equilibrium expression of $\mathcal{V}_{\text{EQ}}(\varphi)$ in Eq. (8) when imposing the detailed balance condition. More importantly, this is consistent with the previous non-equilibrium steady-state definition that was proposed in Eq. (13).

Both routes in Sec. 3.1 and Sec. 3.2 led us to the same definition for the effective mean-field potential $\mathcal{V}_{\text{NESS}}(\varphi)$ in Eqs. (13) and (18). We argue in App. A that the denominator $D(\varphi)$ is inconsequential to the resulting theory close to a continuous phase transition. Furthermore, we show on general grounds in the App. B that the dynamical ratio $\hat{R}/R(\varphi)$ has the following structure around $\varphi \sim 0$,

$$\hat{R}/R(\varphi) \sim \beta_0 \varphi + C_\alpha \operatorname{sgn}(\varphi)|\varphi|^{1+\alpha} + \dots \,, \tag{20}$$

where $\beta_0, \alpha > 0$. This justifies the structure of the effective potential $\mathcal{V}_{\text{NESS}}(\varphi)$ announced in Eq. (11).

### 3.3 Effective temperature

Alternatively to generalizing the definition of the effective potential to non-equilibrium steady states, $\mathcal{V}_{\text{NESS}}(\varphi)$ in Eqs. (13) and (18), one can decide to stick with the equilibrium potential, $\mathcal{V}_{\text{EQ}}(\varphi)$ in Eq. (7) at the cost of absorbing the non-analyticities into a redefinition of the temperature. One can indeed define an order-parameter-dependent effective temperature $T_{\text{eff}}(\varphi) \equiv \beta_{\text{eff}}(\varphi)^{-1}$ by imposing an effective detailed-balance condition, namely

$$\hat{R}/R(\varphi) \equiv \tanh\left(\beta_{\text{eff}}(\varphi)\varphi\right) \,. \tag{21}$$

One obtains the regular $\varphi^4$ potential

$$\mathcal{V}_{\text{NESS}}(\varphi) = \frac{1}{2}[1 - \beta_{\text{eff}}(\varphi)]\varphi^2 + \frac{1}{12}\beta_{\text{eff}}^3(\varphi)\varphi^4 + o(\varphi^4) \,, \tag{22}$$

where $o(\varphi^4)$ stands for terms that are of higher order than $\varphi^4$ and where the effective temperature reads

$$\beta_{\text{eff}}(\varphi) \simeq \beta_0 + \frac{C_\alpha}{2 + \alpha}|\varphi|^\alpha \,, \tag{23}$$

with $\beta_0, \alpha > 0$. This alternate construction is particularly valuable when one has a clear physical understanding of the non-equilibrium processes responsible for the variation of the temperature away from its thermodynamical value. Recently, such a viewpoint was used in a related non-equilibrium steady-state $\mathbb{Z}_2$-symmetry breaking scenario: in the context of the resistive switching of anti-ferromagnetic insulators driven by a DC voltage, where the local heating and $T_{\text{eff}}(\varphi)$ could be computed exactly from first principles [13].

## 4 Concrete examples around the Ising model

Using two concrete examples of driven-dissipative Ising models, one with a DC drive and the other with an AC drive, we shall derive explicitly the non-analytic terms entering the effective potential announced in Eq. (11). They are of the type

$$\mathcal{V}_{\text{NESS}}(\varphi) = \dots + c_\alpha|\varphi|^{2+\alpha} + \dots \,, \tag{24}$$

where $\alpha > 0$ and the coefficient $c_\alpha$ is a smooth function of the external parameters that vanishes at equilibrium.

## 4.1 Dissipative Ising model coupled to multiple baths

Consider the Ising model in Eq. (3) where each spin is now weakly coupled to two independent baths at two different temperatures $T_1$ and $T_2$, and with two system-bath hybridization functions $v_1(\omega)$ and $v_2(\omega)$, respectively. Those correspond to the state broadening, at energy $\omega$, due to the local spin-flip dynamics induced by each bath. Typically, $v_i(\omega) = \gamma_i \rho_i(\omega)$ where $\gamma_i > 0$ is a system-bath coupling constant and $\rho_i(\omega)$ is the density of states of the bath.

**Mean-field Lindblad description**   In the single-spin mean-field approach, the impurity Hamiltonian reads

$$H = -\varphi S^z \,, \tag{25}$$

and the dynamics of the impurity spin density matrix $\rho$ are given (within the regular Born-Markov approximation[2]) by the following Lindblad-type Master equation:

$$\partial_t \rho = -i[H, \rho] + v_1(\epsilon) \left\{ (1 + n_B(\epsilon, T_1)) \mathcal{D}[\sigma_\varphi^+]\rho + n_B(\epsilon, T_1)\mathcal{D}[\sigma_\varphi^-]\rho \right\}$$
$$+ v_2(\epsilon) \left\{ (1 + n_B(\epsilon, T_2)) \mathcal{D}[\sigma_\varphi^+]\rho + n_B(\epsilon, T_2)\mathcal{D}[\sigma_\varphi^-]\rho \right\} \,. \tag{26}$$

Such Lindblad equations are commonly used to describe the dynamics of quantum systems weakly coupled to an environment. They characterize the non-unitary evolution of the system's reduced density matrix, *i.e.* once the degrees of freedom of the environment have been traced out. They can be derived unambiguously under the Born-Markov approximation (*i.e.* within the same regime of validity as the Fermi Golden rule), assuming the immediate environment is relaxing sufficiently fast to remain unaffected by the state of the system. In the limit of no environment, they reduce to the usual von Neumann equation of isolated dynamics, see the first term in the RHS of Eq. (26). The following terms in the RHS originate from the hybridization with the environment. They involve the quantum of energy exchanged with the environment $\epsilon \equiv 2|\varphi|$, the Bose-Einstein distribution $n_B(\omega, T) \equiv 1/(e^{\omega/T} - 1)$, the jump operators $\sigma_\varphi^\pm \equiv S^\pm$ when $\varphi > 0$ and $\sigma_\varphi^\pm \equiv S^\mp$ when $\varphi < 0$, and the Lindblad operators $\mathcal{D}[X]\rho \equiv X\rho X^\dagger - (X^\dagger X\rho + \rho X^\dagger X)/2$ that act linearly on the system's density matrix, while preserving its trace, hermiticity, and positivity. It is rather straightforward to show that

$$\hat{R}/R(\varphi) = \frac{v_1(2|\varphi|) + v_2(2|\varphi|)}{v_1(2|\varphi|)\coth(\varphi/T_1) + v_2(2|\varphi|)\coth(\varphi/T_2)} \,. \tag{27}$$

The presence of the absolute values $|\varphi|$ in the arguments of the bath hybridization functions is a first possible source of non-analyticites in $\hat{R}/R(\varphi)$. A second source of non-analyticities is a non-integer power law of the low-energy spectrum of the bath hybridization functions, *i.e.* $v_i(\omega) \sim \omega^\alpha$ with $\alpha \notin \mathbb{N}$.

Note that any possible non-analyticity in the ratio $\hat{R}/R(\varphi)$ is lost as soon as the two bath hybridization functions behave identically, *i.e.* $v_1(\omega) \propto v_2(\omega)$, since they can be factored out of the expression (27). Incidentally, the continuous ferromagnetic transition in this subclass of non-equilibrium models, often investigated under the name of "competing spin-flip dynamics", was repeatedly found to belong to the equilibrium Ising universality class [14–16]. Another trivial case controlled by the Ising universality class is the thermal equilibrium limit, *i.e.* $T_1 = T_2$, in which detailed balance and the analyticity of $\hat{R}/R(\varphi)$ are naturally recovered.

---

[2]Note that the Markov approximation behind the derivation of the above Lindblad-Master equation becomes exact once a steady state is reached.

We now stay away from these special cases, and assume that the low-energy features of the baths are such that $\nu_1(\omega) \gg \nu_2(\omega)$. Eq. (27) yields the self-consistency equation

$$\varphi \overset{\text{SC}}{=} \hat{R}/R(\varphi) \simeq \tanh(\varphi/T_1) \left[ 1 + \frac{\nu_2(2|\varphi|)}{\nu_1(2|\varphi|)} \left( 1 - \frac{\tanh(\varphi/T_1)}{\tanh(\varphi/T_2)} \right) \right] + \dots \tag{28}$$

$$\overset{\varphi \sim 0}{\simeq} \frac{\varphi}{T_1} \Big[ 1 + \underbrace{c_{21} \left( 1 - \frac{T_2}{T_1} \right) |\varphi|^{\alpha_{21}}}_{\text{non-analytic}} - \frac{1}{3} \left( \frac{\varphi}{T_1} \right)^2 \Big] + \dots , \tag{29}$$

where we assumed the low-energy power-law behaviors $\nu_i(2\omega) \simeq c_i \, \omega^{\alpha_i}$ and introduced $\alpha_{21} \equiv \alpha_2 - \alpha_1$ and $c_{21} \equiv c_2/c_1$.

**Effective potential**  Using the definition in Eqs. (13) or (18) with $D(\varphi) := 1$, we obtain the following effective potential

$$\mathcal{V}_{\text{NESS}}(\varphi) = \frac{1}{2}(1 - \beta_1)\varphi^2 - \frac{c_{21}}{2 + \alpha_{21}}\beta_1(1 - \beta_1/\beta_2)|\varphi|^{2+\alpha_{21}} + \frac{1}{12}\beta_1^3\varphi^4 . \tag{30}$$

Let us use this example to underline once again the main message of this manuscript. We have derived a non-equilibrium effective potential for the non-equilibrium steady states, which is $\mathbb{Z}_2$-symmetric, but not an analytic function of $\varphi$. The quadratic and quartic term are analytic, and their prefactors are smooth functions of the external parameters. It is the non-equilibrium nature of the environment which is responsible for the non-analytic term in $|\varphi|^{2+\alpha_{21}}$. The prefactor of the latter is a smooth function of the external parameters that vanishes at equilibrium (when $T_1 = T_2$). The exponent $\alpha_{21} > 0$ can be non-integer valued. For example, for $d$-dimensional baths with dispersion relations $\omega \sim k^z$, the exponent $\alpha_{21} = d_2/z_2 - d_1/z_1$ is a rational number.

An equivalent description of the physics consists in sticking to the equilibrium $\varphi^4$ potential, in exchange of working with the effective temperature

$$T_{\text{eff}}(\varphi) = T_1 - \frac{2c_{21}}{2 + \alpha_{21}}(T_1 - T_2)|\varphi|^{\alpha_{12}} . \tag{31}$$

**Phase transition**  The effective potential in Eq. (30) reveals a continuous phase transition at the critical temperature $T_1^c = 1$ as long as the second bath is at a higher temperature than the first, *i.e.* for any $T_2 \geq T_1$. Noteworthy, this critical temperature is as if the system was only coupled, and in equilibrium, with the first bath: The first bath is the most "relevant" bath. However, the mean-field critical exponents are clearly modified with respect to their equilibrium values. If $\alpha_{21} < 2$, the non-analytic term in Eq. (29) dominates over $\varphi^4$ term at small $\varphi$, and we get the scaling law

$$|\varphi| \sim \left( \frac{\tau_1}{\tau_1 - \tau_2} \right)^{\hat{\beta}_{\text{NESS}}} , \tag{32}$$

where we introduced the reduced temperatures $\tau_i \equiv 1 - T_i$ and the mean-field critical exponent

$$\hat{\beta}_{\text{NESS}} = \frac{1}{\alpha_{21}} . \tag{33}$$

This critical exponent is much different, in origin and in value, from its equilibrium counterpart $\hat{\beta}_{\text{EQ}} = 1/2$ which stems from the competition of the $\varphi^2$ and the $\varphi^4$ terms of the mean-field potential. If $1 < \alpha_{21} < 21$, the continuous phase transition can be classified as a third-order phase transition since the derivative of the order parameter is continuous across the transition,

whereas $0 \leq \alpha_{21} \leq 1$ yields a second-order phase transition with a discontinuous derivative across the transition.

Remarkably, this continuous phase transition disappears if the second bath is colder than the first, *i.e.* $T_2 < T_1$. There, we rather get a discontinuous (first-order) phase transition at a different critical temperature $T_2^c(T_1)$.

**Many-bath dissipative Ising model** The previous discussion can be generalized when the Ising spins are coupled to more than two baths. Considering multiple baths, indexed by $n = 1, 2, \ldots$, with different[3] temperature $T_n$, chemical potential $\mu_n$ and hybridization function $\nu_n(\omega) \sim \omega^{\alpha_n}$ at low energies, the ratio in Eq. (27) simply generalizes to

$$\hat{R}/R(\varphi) = \frac{\sum_n \nu_n(2|\varphi|)}{\sum_n \nu_n(2|\varphi|) \coth(\frac{2\varphi - \text{sgn}(\varphi)\mu_n}{2T_n})} . \tag{34}$$

Assuming that $\nu_1(\omega) \gg \nu_2(\omega) \gg \nu_3(\omega) \gg \ldots$ at low energies, we may neglect the baths indexed by $n \geq 3$ and the situation boils down to the previous case of two independent baths, yielding the critical exponent $\hat{\beta}_{\text{NESS}}$ already computed in Eq. (33).

Importantly, this teaches us that criticality is controlled in the non-equilibrium steady states by those two baths that have the largest hybridization functions (*i.e.* typically the largest density of states) at low energies.

## 4.2 Floquet-driven dissipative Ising model

In this example, we borrow the driven-dissipative model studied in [17]. It consists of the dissipative Ising model, weakly coupled to a thermal bath, and driven out of equilibrium by a periodic longitudinal field with frequency $\Omega$ and amplitude $h \geq 0$. The time-dependent Hamiltonian reads

$$H(t) = -\frac{1}{z} \sum_{\langle ij \rangle} S_i^z S_j^z + \sum_i h \cos(\Omega t) S_i^z . \tag{35}$$

By combining the standard single-spin mean-field approximation with a Floquet treatment of the periodic drive, one derives the steady-state dynamics of the order parameter averaged over one period $2\pi/\Omega$, $\varphi \equiv \overline{\langle S_i^z \rangle}$. In the regime where $\Omega > 2|\varphi|$, one obtains

$$\hat{R}/R(\varphi) = \frac{J_0^2 \nu(2|\varphi|) + \sum\limits_{m \in \mathbb{Z}^*} \text{sgn}(m) J_m^2 \nu(|m|\Omega + \text{sgn}(m)2|\varphi|)}{J_0^2 \nu(2|\varphi|) \coth(\frac{\varphi}{T}) + \sum\limits_{m \in \mathbb{Z}^*} \text{sgn}(m) J_m^2 \nu(|m|\Omega + \text{sgn}(m)2|\varphi|) \coth\left(\frac{2\varphi + \text{sgn}(\varphi)m\Omega}{2T}\right)}, \tag{36}$$

where $\nu(\omega)$ is the hybridization function with the bath, and $J_m \equiv J_m(2h/\Omega)$ where $J_m(x)$, $m \in \mathbb{Z}$, are the Bessel functions of the first kind.

We now make a connection with the Sec. 4.1 by recasting the impurity problem at hand into an impurity spin coupled to multiple equilibrium baths. Such a decomposition of a given non-equilibrium impurity environment into a collection of equilibrium baths has already been made in the context of non-equilibrium dynamical mean-field theory [18]. Here, the expression of the ratio $\hat{R}/R(\varphi)$ in Eq. (36) can be formally recast in the form of Eq. (34) by identifying the

---

[3]If two (or more) baths have the same temperature and chemical potential, they can be formally combined into a single bath.

following equilibrium baths

$$m = 0 : \begin{cases} \nu_0(\omega) & := J_0^2 \, \nu(\omega) \\ T_0 & := T \\ \mu_0 & := 0 \end{cases} \qquad m \neq 0 : \begin{cases} \nu_m(\omega) & := \text{sgn}(m) J_m^2 \, \nu(|m|\Omega + \text{sgn}(m)\omega) \\ T_m & := T \\ \mu_m & := -m\Omega \end{cases} ,$$

(37)

with $m \in \mathbb{Z}$ and where $\nu_m$, $T_m$, and $\mu_m$ are the $m^{\text{th}}$ bath hybridization function, temperature, and chemical potential, respectively. Note that the sign of $\nu_m(\omega)$ may not be positive in this Floquet approach.

**Strong-driving regime** In the strong-driving regime where $\Omega, h \gg |\varphi|, T$, this further simplifies as

$$\hat{R}/R(\varphi) \simeq \frac{J_0^2 \, \nu(2|\varphi|) + 2|\varphi|A}{J_0^2 \, \nu(2|\varphi|) \coth(\varphi/T) + 2 \, \text{sgn}(\varphi) B} , \tag{38}$$

where we introduced $A \equiv 2 \sum_{n>0} J_n^2 \, \nu'(n\Omega)$ and $B \equiv \sum_{n>0} J_n^2 \, \nu(n\Omega) \geq 0$. Assuming a power-law behavior of the low-energy spectrum of the bath hybridization function, *i.e.* $\nu(\omega) \simeq c \, \omega^\alpha$ with $\alpha > 0$ and $c > 0$, we get

$$\hat{R}/R(\varphi) \overset{\varphi \sim 0}{\simeq} \beta_0 \, \varphi + C_\alpha \, \text{sgn}(\varphi)|\varphi|^{1+|\alpha-1|} + \dots , \tag{39}$$

where the symbol $\dots$ here stands for a collection of higher-order terms of the form $|\varphi|^{1+n|\alpha-1|}$ with $n \geq 2$. The coefficients

$$\beta_0 = \begin{cases} \beta \\ \beta \frac{1+\beta_\Omega \epsilon_0}{1+\beta \epsilon_0} \\ \beta_\Omega \end{cases} , \qquad C_\alpha = 2^{|\alpha-1|} \times \begin{cases} \beta \, \epsilon_0 \, (\beta_\Omega - \beta) & \alpha < 1 \\ 0 & \alpha = 1 \\ \frac{1}{\beta \, \epsilon_0} (\beta - \beta_\Omega) & \alpha > 1 \end{cases} , \tag{40}$$

where $\beta_\Omega \equiv A/B \geq 0$ and $\epsilon_0 \equiv B/cJ_0^2 > 0$, smoothly depend on the external parameters such as the temperature $T$, the driving amplitude $h$, or the driving frequency $\Omega$ [4]. Importantly, we find that both cases $\alpha < 1$ and $\alpha > 1$ give rise to non-analytic terms that enter the expression of $\hat{R}/R(\varphi)$ above the first order in $\varphi$. The case of an Ohmic bath, *i.e.* $\alpha = 1$, is special because $\hat{R}/R(\varphi)$ is analytic in $\varphi$ and we recover equilibrium physics at a modified temperature. Incidentally, the Ohmic case has been explored numerically in 2D [19,20] and in 3D [21], and was indeed found to belong to the Ising universality class. However, to the best of our knowledge, the generic case of a non-Ohmic bath has not been studied.

Ultimately, this yields the following structure of the effective potential,

$$\mathcal{V}_{\text{NESS}}(\varphi) = \frac{1}{2}(1 - \beta_0)\varphi^2 - \frac{C_\alpha}{2 + |\alpha-1|}|\varphi|^{2+|\alpha-1|} + \dots . \tag{41}$$

Equivalently, this corresponds to an order-parameter-dependent effective (inverse) temperature reading

$$\beta_{\text{eff}}(\varphi) := \beta_0 + \frac{C_\alpha}{2 + |\alpha-1|}|\varphi|^{|\alpha-1|} + \dots . \tag{42}$$

Using the results of Sec. (4.1), this predicts a continuous non-equilibrium phase transition at the bath critical temperature $T^c = 1$ whenever $B/A > 1$ in the sub-Ohmic case ($\alpha < 1$), and at the critical drive $B/A = 1$ whenever $T > 1$ in the super-Ohmic case ($\alpha > 1$). Both these transitions are described by a mean-field critical exponent $\hat{\beta}_{\text{NESS}} = 1/|\alpha-1|$.

---

[4]The equilibrium limit cannot be easily recovered since we have assumed strong-driving conditions, $\Omega, h \gg |\varphi|, T$.

# 5 Beyond mean field – discussion and open problems

In this Section, we question the lessons of the previous mean-field analysis away from the limit of infinite dimensionality, and we propose an effective Landau-Ginzburg free energy for the non-equilibrium steady states in finite dimensions.

**Non-analyticity vs. discreteness**   The non-analytic term of the effective potential originated from the ratio $\hat{R}/R(\varphi)$ when evaluated around $\varphi \sim 0$. While there is no question that this function of $\varphi$ can feature non-analyticities in a non-equilibrium steady state (we have computed it explicitly in a couple of concrete examples), the fact that it was continuously probed around $\varphi \sim 0$ was clearly due to the self-consistency equation of the mean-field treatment, namely $\varphi \overset{\text{SC}}{=} \hat{R}/R(\varphi)$. In practice, the dynamics of any single spin depends of the local Weiss field $h_W = \frac{n_\uparrow - n_\downarrow}{z}$ where $n_\uparrow$ ($n_\downarrow$) counts the number of up (down) spin neighbors and $z = n_\uparrow + n_\downarrow$ is the coordination number. In finite dimensions, $h_W$ is not a continuous variable but a discrete quantity which varies by increments of $\delta \equiv 2/z$. This implies that the dynamics of a given spin is controlled by the discrete set of values $\hat{R}/R(n\delta)$, $n \in \mathbb{Z}$ rather than by the continuous series expansion of $\hat{R}/R(\varphi)$ around $\varphi \sim 0$. Therefore, it is legitimate to worry whether the non-analyticities of $\hat{R}/R(\varphi)$ are still transfered to $\mathcal{V}_{\text{NESS}}(\varphi)$ in finite dimensions, or if they are washed away with the introduction of a small energy cutoff in the theory.

**Coarse-graining and Landau-Ginzburg free-energy**   The above issue could in principle be removed by performing a coarse-graining procedure, where the size of the coarse-graining region would replace the connectivity $z$ of the lattice. In this case, the discreteness of the Weiss field would be exactly the same as the one of the coarse-grained magnetization $\varphi(x)$, which in Landau theory is replaced by a continuous field. If one assumes that there exists an appropriate coarse-graining procedure that allows to neglect the energy discretization along with the order parameter discretization, then one can extend the equilibrium reasoning that led to Eq. (9) to propose a Landau-Ginzburg free energy of the form:

$$\mathcal{F}_{\text{NESS}}[\varphi] = \int \mathrm{d}^d x \left\{ \frac{1}{2}(\nabla\varphi)^2 + \int^{\varphi(x)} \mathrm{d}\varphi \, \left[ \varphi - \hat{R}/R(\varphi) \right] \right\} \tag{43}$$

$$\simeq \int \mathrm{d}^d x \, \frac{1}{2}(\nabla\varphi)^2 + a_2 \varphi^2 + c_\alpha |\varphi|^{2+\alpha} + a_4 \varphi^4 \,, \tag{44}$$

where non-analytic terms can enter the expression at the order $|\varphi|^{2+\alpha}$ and the exponent $\alpha > 0$ is typically determined by law-energy spectrum of the environment. The parameter $c_\alpha$ is a smooth function of the external parameters (temperatures, driving strength, etc), that vanishes at equilibrium thus restoring the analyticity of the free-energy density.

The presence of analytic gradient terms in the Landau-Ginzburg free energy, such as $(\nabla\varphi)^2$ in Eq. (43), is guided by what is already well-known in equilbrum. Depending on the physics at stake, terms with higher order (integer) derivatives, higher (integer) powers, or mixed terms such as $\varphi^2(\nabla\varphi)^2$ could naturally enter the free energy. In the context of the $\varphi^4$ magnet, we know that they are essentially irrelevant in front of the $(\nabla\varphi)^2$ term. However, in a generic non-equilibrium steady state, we cannot rule out the additional presence of non-analytic terms of the form $|\varphi|^\alpha(\nabla\varphi)^2$, with a non-integer power $\alpha > 0$ [5]. While a naive power counting seems to indicate that it would be less relevant than the original $(\nabla\varphi)^2$ term, more insight is required (*e.g.* numerical simulations or full-fledged RG analysis) to make a definite statement.

---

[5]We did not propose a term of the form $|\nabla\varphi|^\alpha$ with a non-integer power $\alpha > 0$ because the non-equilibrium environments considered in this manuscript are local in space, and we do not expect them to generate gradient terms.

**RG approach**    If the assumptions leading to this Landau-Ginzburg free energy are valid, then the question becomes how information can be extracted from it.

The first step is to analyze the engineering dimension of the non-analytic term. Setting as usual the dimension of the gradient term to be zero, we get $[c_\alpha] = \alpha(d/2-1)-2$. For $\alpha \in (0,2)$, this term is always more relevant than the $\varphi^4$ term. Above $d > 4/\alpha + 2$, the non-analytical term is irrelevant. Therefore, to the extent that one can carry this naive analysis of scaling dimensions, one would expect that new mean-field exponents obtained in the preceding part of the paper would apply in high enough dimensions.

However, going beyond the tree-level power counting is daunting. The following two issues with a proper RG calculation arise. First, the presence of the non-analytical potential makes it difficult to carry out a conventional RG diagrammatic calculation. (Possibly, a functional RG approach may be better suited instead.) Second, it is possible that the presence of the non-analytic potential at tree level may be symptomatic that a proper RG scheme should not start with it, but instead take a step back and restore the bath degrees of freedom instead of integrating them out to get the effective potential.

**Monte-Carlo approach**    We would be cautious in diving into an RG calculation with the Landau-Ginzburg free energy Eq. (43) before we could more solidly establish the validity of the assumption that coarse-graining resolves the issue of non-analyticity vs. discreteness discussed above. That could be settled by numerical simulations of the lattice model in Sec. 4.1 in 2D and 3D. This should validate or invalidate that the critical exponents (as well as the order of the transition) at the magnetic transition acquire a dependency on the bath density of states. If so, this would provide solid evidence for remnants of universality in non-equilibrium steady states. If not, we still expect the presence of a near-critical crossover regime controlled by these bath-dependent exponents.

# Acknowledgements

C. C. thanks the hospitality of the École Normale Supérieure in Paris, where this work was initiated. We are grateful to L. Cugliandolo for pointing out relevant literature.

**Funding information**    This work benefited from the support of the ANR project MOMA (C. A.) and from the DOE Grant No. DE-FG02-06ER46316 (C. C.).

# A   Inconsequentiality of the denominator $D(\varphi)$

Let us recall the definition of the effective potential that we proposed from a single-spin mean-field treatment of the dynamics of the dissipative Ising model:

$$\mathcal{V}(\varphi) \equiv \int^{\varphi} \mathrm{d}\varphi \; \frac{\varphi - \hat{R}/R(\varphi)}{D(\varphi)} \; . \tag{45}$$

$\mathcal{V}(\varphi)$ is such that its extrema, located at $\partial_\varphi \mathcal{V} = 0$, are in one-to-one correspondence with the solutions of the self-consistency equation $\varphi \overset{\text{SC}}{=} \hat{R}/R(\varphi)$.

The possible expressions of $D(\varphi)$ are subject to the following constraints:

- Symmetry: the $\mathbb{Z}_2$ symmetry of the effective potential imposes $D(\varphi)$ to be even, *i.e.* $D(\varphi) = D(-\varphi)$,

- Stability of the solutions: $D(\varphi)$ is non-negative, *i.e.* $D(\varphi) \geq 0$.

- Stability of the theory: $D(\varphi)$ cannot affect the positive sign of the coefficient of the highest relevant power of $\varphi$ in $\mathcal{V}(\varphi)$

- Re-parametrization invariance: at criticality, all possible expressions of the effective potential should match, *i.e.* $D(\varphi = 0) = 1$.

- "Universality" of the definition: $V(\varphi)$ and therefore $D(\varphi)$ are expected to be functions of $\varphi$ and $\hat{R}/R(\varphi)$ which do not depend explicitly on the system parameters. In particular, this implies that the expression of $D(\varphi, \hat{R}/R)$ is the same at equilibrium and out of equilibrium, and is analytic in both its arguments: $D(\varphi) = f(\varphi^2, \varphi\hat{R}/R, (\hat{R}/R)^2, \ldots)$

- RG relevance: the effective potential being defined up to terms which are irrelevant with respect to a $\varphi^4$ interaction, one may truncate $D(\varphi)$ at the order $\varphi^2$.

Thus, a given choice of $D(\varphi)$ may only impact the precise form of the potential by modifying the value (not the sign) of the coefficient of the $\varphi^4$ term. Given that the parameters of the Landau-Ginzburg free-energy density are anyway immaterial, we can conclude that $D(\varphi)$ is essentially inconsequential close to a continuous phase transition.

# B  Structure of the dynamical ratio $\hat{R}/R(\varphi)$

We consider the single-spin mean-field impurity problem associated with a driven-dissipative Ising model that has reached a non-equilibrium steady state. It consists of a single spin coupled to a local Weiss field and to an incoherent environment responsible for spin flips at rates $R_{\uparrow\downarrow}$ and $R_{\downarrow\uparrow}$. We do not assume the environment to be at equilibrium, *i.e.* the rates do not have to obey the detailed balance condition.

We recall the self-consistency equation on the mean-field order parameter,

$$\varphi \overset{\mathrm{SC}}{=} \hat{R}/R(\varphi)\,, \tag{46}$$

where we introduced $\hat{R} \equiv R_{\downarrow\uparrow} - R_{\uparrow\downarrow}$ and $R \equiv R_{\downarrow\uparrow} + R_{\uparrow\downarrow}$.

Below we assume

- The existence of a continuous phase transition.

- Smoothness of the rates: $R_{\downarrow\uparrow}$ and $R_{\uparrow\downarrow}$ are smooth continuous functions of the external parameters (temperature, drive, etc.)

The ratio $\hat{R}/R(\varphi)$ is a continuous and odd function of $\varphi$. This guarantees that the paramagnetic $\varphi_{\mathrm{PM}} = 0$ is always a solution of the self-consistency equation. If any, the other solutions are ferromagnetic and obey

$$1 \overset{\mathrm{SC}}{=} \frac{\hat{R}/R(\varphi_{\mathrm{FM}})}{\varphi_{\mathrm{FM}}}\,. \tag{47}$$

Close to a continuous phase transition, the two solutions merge together, *i.e.* $\varphi_{\mathrm{FM}} \to \varphi_{\mathrm{PM}} = 0$. This imposes that, at criticality,

$$\lim_{\varphi \to 0} \frac{\hat{R}/R(\varphi)}{\varphi}\bigg|_{\mathrm{criticality}} = 1\,. \tag{48}$$

Using the assumption that the rates are smooth continuous functions of the external parameters, we can therefore conclude that away from criticality

$$\lim_{\varphi \to 0} \frac{\hat{R}/R(\varphi)}{\varphi} = \beta_0 \,, \tag{49}$$

where the constant $\beta_0 > 0$ depends smoothly on the external parameters and $\beta_0 = 1$ at criticality. The property in Eq. (49) ensures that the development of $\hat{R}/R(\varphi)$ in powers of $\varphi$ starts at the order $\varphi$:

$$\hat{R}/R(\varphi) = \beta_0 \varphi + \dots \,, \tag{50}$$

with $\beta_0 > 0$. Importantly, this allows the presence of non-analytic terms of the type $\mathrm{sgn}(\varphi)|\varphi|^{1+\alpha}$ as long as $\alpha > 0$.

## C    Fokker-Planck equation on $P(\varphi, t)$

We work with the fully-connected version of the driven-dissipative Ising model that we considered in Sec. 3.2. We recall the system Hamiltonian given in Eq. (51)

$$H = -\frac{1}{N} \sum_{ij} S_i^z S_j^z \,, \tag{51}$$

where the total number of spins $N$ is large but finite. We assume that the non-equilibrium environment is uniform and acts locally on the spins. We follow the dynamics of the mean magnetization,

$$\varphi \equiv \frac{1}{N} \sum_{i=1}^{N} S_i^z \,. \tag{52}$$

It is a stochastic process in which the random jumps are due to individual spin flips driven by the system-bath interaction. They occur in units of $\delta\varphi = \frac{2}{N}$. In a mean-field picture, the spin flips are supposed uncorrelated but their statistics, namely the spin flip rates $R_{\downarrow\uparrow}$ and $R_{\uparrow\downarrow}$, are controlled by the common mean-field order parameter $\varphi$. In this approximation, the probabilities for $\varphi$ to increase by $\delta\varphi$, decrease by $\delta\varphi$, or stay constant, during a time step $\mathrm{d}t$ read, respectively

$$\begin{cases} R_+(\varphi \mapsto \varphi + \delta\varphi) &= \underbrace{N P_\downarrow R_{\downarrow\uparrow}}_{\equiv r_+} \mathrm{d}t \\ R_-(\varphi \mapsto \varphi - \delta\varphi) &= \underbrace{N P_\uparrow R_{\uparrow\downarrow}}_{\equiv r_-} \mathrm{d}t \\ R(\varphi \mapsto \varphi) &= 1 - (r_+ + r_-)\,\mathrm{d}t \end{cases} \,, \tag{53}$$

where $P_\downarrow = \frac{1-\varphi}{2}$ ($P_\uparrow = \frac{1+\varphi}{2}$) is the probability for a given spin to be down (up), $R_{\downarrow\uparrow}$ ($R_{\uparrow\downarrow}$) is the rate of flipping a spin up (down), and the overall factor of $N$ takes care of the summation over the $N$ (uncorrelated) spins in Eq. (52).

   The derivation of the Fokker Planck equation which governs the dynamics of the probability to find $\varphi$ at time $t$, $P(\varphi, t)$, is obtained quite standardly by developing

$$P(\varphi, t + \mathrm{d}t) \simeq P(\varphi, t) + \mathrm{d}t \, \partial_t P(\varphi, t) + \mathcal{O}(\mathrm{d}t^2) \tag{54}$$

on the one side, and by developing the budget equation

$$
\begin{aligned}
P(\varphi, t + \mathrm{d}t) =& P(\varphi - \delta\varphi, t)R_+(\varphi - \delta\varphi \mapsto \varphi) \\
&+ P(\varphi + \delta\varphi, t)R_-(\varphi + \delta\varphi \mapsto \varphi) \\
&+ P(\varphi, t)R(\varphi \mapsto \varphi)
\end{aligned}
\tag{55}
$$

$$
\begin{aligned}
\simeq & [Pr_+ - \delta\varphi\, \partial_\varphi(Pr_+) + \tfrac{1}{2}\delta\varphi^2\, \partial_\varphi^2(Pr_+)]\mathrm{d}t \\
&+ [Pr_- + \delta\varphi\, \partial_\varphi(Pr_-) + \tfrac{1}{2}\delta\varphi^2\, \partial_\varphi^2(Pr_-)]\mathrm{d}t \\
&+ P[1 - (r_+ + r_-)\mathrm{d}t] + \mathcal{O}(\delta\varphi^3)
\end{aligned}
\tag{56}
$$

on the other side. Finally, identifying Eqs. (54) and (55), we obtain the Fokker Planck equation

$$
\begin{aligned}
\partial_t P =& -\partial_\varphi\left(P[(R_{\downarrow\uparrow} - R_{\uparrow\downarrow}) - \varphi(R_{\downarrow\uparrow} + R_{\uparrow\downarrow})]\right) \\
&+ \frac{1}{N}\partial_\varphi^2\left(P[(R_{\downarrow\uparrow} + R_{\uparrow\downarrow}) - \varphi(R_{\downarrow\uparrow} - R_{\uparrow\downarrow})]\right) + \mathcal{O}(1/N^2),
\end{aligned}
\tag{57}
$$

which can be recast into a simple conservation equation,

$$
\partial_t P + \partial_\varphi J = 0,
\tag{58}
$$

via the identification of the current density

$$
J \equiv PR(\hat{R}/R - \varphi) - \frac{1}{N}\partial_\varphi\left[PR(1 - \varphi\hat{R}/R)\right] + \mathcal{O}(1/N^2),
\tag{59}
$$

with $R \equiv R_{\downarrow\uparrow} + R_{\uparrow\downarrow}$ and $\hat{R} \equiv R_{\downarrow\uparrow} - R_{\uparrow\downarrow}$.

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
