# Peer review of "Landau Theory for Non-Equilibrium Steady States"

_SciPost Physics, doi:SciPost Phys. 8, 074 (2020)_

## Round 1 · Referee Report · Anonymous · 2019-10-29

Report

Report on: Landau Theory for Non-Equilibrium Steady States
by C. Aron and C. Chamon

This paper by Aron and Chamon addresses the field theoretical
description of several cases of driven-dissipative (spin)
systems. Different types of non-equilibrium are considered, such as
coupling to multiple heat baths which have different temperatures and
also driving via a time-periodic external magnetic field. Both cases
are generic and there is much interest in studying the ensuing
non-equilibrium properties of systems with large numbers of coupled
degrees of freedom.

In the paper the authors show how one can derive a corresponding free
energy functional from the (standard) self-consistency condition of
the behaviour of a single spin in an external field and the mean
magnetization of the system. Their treatment applies close to a
continuous phase transition and the procedure is clearly explained in
equilibrium and then generalized and used in nonequilibrium. Both
"bottom-up" and "top-down" approaches are used in order to obtain the
functional form the resulting Landau potential.

The authors argue and then for specific cases explicitly derive the
corrections to the standard phi^4 free energy and show that these
additional terms feature non-analytical power-law dependence on the
order parameter. A concluding section contains an enlightening
discussion of various substantial points.

The paper contains high quality theoretical work and I recommend
publication in SciPost Physics. I summarize three main points and
several minor points below and leave it to the authors to address
these in order to potentially improve their paper further.

1) A bit more detail around the Lindblad equation (26) and solution
(27) would help readability. I guess many readers who have been
familiar with the material presented so far will struggle at this
point. I would hope that giving a little more background and
explanations could help (without having to turn this section into a
full-blown tutorial of dissipative quantum dynamics).

2) Could the gradient terms (Sec.5) not also feature new,
non-analytical contributions?

3) Would one need additional order parameters in order to describe
time-dependent nonequilibrium?

Minor points are the following.

p.7 "The equations (4) and (5) still apply to a non-equilibrium
scenario." Concerning (5), I guess this applies only to the first
equality. Maybe specify.

p.7 I guess "fully-connected" refers to the sums in (14) running over
all pairs, not just next neighbours. Maybe spell this out to avoid
any uncertainty.

Beginning of Sec.4.1. Define variable omega. A bit of description of
the concept of the hybridization functions would help

Typos: class of of, an homogeneous, can indeed defined.

---

## Round 2 · Author Response

We are grateful to the Referee for his/her in-depth review of the Manuscript. We are pleased that our core results reached him/her in a clear way. The Referee is making pertinent comments and asking insightful questions. We address some of them, while some others go beyond the scope of this first manuscript, and frankly, beyond our current understanding of the full theory we aspire at developing. We included the latter in the last section, "Beyond mean field -- discussion and open problems", and are committed to addressing them in future works.

1) Following the Referee's suggestion, we included a paragraph to give a little more background to the Lindblad equation that is employed to describe the dissipative quantum dynamics. We reproduce this paragraph here below: "Such Lindblad equations are commonly used to describe the dynamics of quantum systems weakly coupled to an environment. They characterize the non-unitary evolution of the system's reduced density matrix, \textit{i.e.} once the degrees of freedom of the environment have been traced out. They can be derived unambiguously under the Born-Markov approximation (\textit{i.e.} within the same regime of validity as the Fermi Golden rule), assuming the immediate environment is relaxing sufficiently fast to remain unaffected by the state of the system. In the limit of no environment, they reduce to the usual von Neumann equation of isolated dynamics, see the first term in the RHS of Eq. (26). The following terms in the RHS originate from the hybridization with the environment. They involve the quantum of energy exchanged with the environment $\epsilon \equiv 2 |\varphi|$, the Bose-Einstein distribution $n_{\rm B}(\omega, T) \equiv 1/(\rme^{\omega/T}-1) $, the jump operators $\sigma_\varphi^\pm \equiv S^\pm$ when $\varphi > 0$ and $\sigma_\varphi^\pm \equiv S^\mp$ when $\varphi < 0$, and the Lindblad operators $\mathcal{D}[X] \rho \equiv X \rho X^\dagger - (X^\dagger X \rho + \rho X^\dagger X)/2$ that act linearly on the system's density matrix, while preserving its trace, hermiticity, and positivity."

2) Yes indeed, in addition to the non-analytic terms of the Landau potential, the field theory for non-equilibrium steady states could also feature non-analytic terms in the gradient terms. They would be of intrinsic non-equilibrium nature. This question can only be answered rigorously by going beyond the mean-field picture, and by accumulating more knowledge (in particular from brute-force numerical simulations) on the generic properties of the statics of these non-equilibrium steady states.

However, given that we are only working with \emph{local} environments, our initial intuition is that tracing out those degrees of freedom should not contribute explicitly to specific non-equilibrium gradient terms in the field theory.

Having said that, real-space coarse-graining could generate non-analytic gradient terms out of those already present in the potential. In the lack of a full-fledged RG computation to provide a clear-cut answer, we may venture into a simple argument:

  • the mean-field approach teaches us that non-analytic terms of the form $|\varphi|^(2+\alpha)$, with $\alpha > 0$, can show up in the Landau potential

  • we saw in Sect. 3.3 that these terms could be equivalently absorbed in the definition of a field-dependent, non-analytic, effective temperature $T_{\rm eff}(\varphi)$

  • in the event that this effective temperature has any physical meaning beyond its mere definition, it would be natural to envision gradient terms of the form $(\nabla \varphi)^2 / T_{\rm eff}(\varphi)$ (this is motivated, \textit{e.g.}, by the Hubbard-Stratonovich construction of the equilibrium Landau-Ginzburg free-energy which features terms of the form $(\nabla \varphi)^2 / T$). In turn, these would yield non-analytic terms of the form $(\nabla \varphi)^2 |\varphi|^{\alpha}$. Simple power counting seems to indicate that they are less relevant than the initial $(\nabla \varphi)^2$ terms (in an RG sense that still has to be sharpened of course).

We included elements of this discussion in Sect. 5 "Beyond mean field -- discussion and open problems", below the free energy proposed in Eq.~(43).

3) "time-dependent nonequilibrium" can encompass different possible scenarios. While the discussion about generic time-dependent many-body states remains equivocal, we may discuss a couple of simpler scenarios that we feel are relevant.

(a) Slowly-varying non-equilibrium states. Assuming that the relaxation mechanisms are effective enough to allow the system to follow the instantaneous steady state set by the interplay of the drive and the dissipation, we naturally expect that a single-order-parameter description is still valid, simply upgrading $\phi(x)$ to $\phi(x,t)$.

(b) Floquet-type scenarios (i.e. with periodic driving). Here, we indeed expect the need for additional order parameters to describe the periodic steady state. While in this manuscript (e.g. in Sect. 4.2 "Floquet-driven dissipative Ising model") we are only discussing the DC component, i.e. the "zero frequency baseline", we expect that the amplitude of each harmonic $n$ of the periodic steady state should also be described by an order parameter $\phi_n(x)$. There, the field theory would therefore take the form of a Landau-Ginzburg Hamiltonian coupling all the field harmonics. Ultimately, an RG analysis could provide insight into the relevance of the different order parameters.

We also thank the Referee for pointing out the typos and help us clarify the few minor points he/she listed. We modified our manuscript accordingly.

---

## Round 2 · List of Changes

• New paragraph below Eq. (26) to give a rapid introduction to Lindblad equations
  • New discussion in Section 5 "Beyond mean field -- discussion and open problems" on the possibility of the presence of non-analytic gradient terms in the Landau-Ginzburg free energy
  • typos and small clarifications here and there

---

## Editorial Decision

published